# Can LLMs Verify Arabic Claims? Evaluating the Arabic Fact-Checking Abilities of Multilingual LLMs

**Ayushman Gupta**\*, **Aryan Singhal**\*, **Thomas Law**\*, **Veekshith Rao**\*,
**Evan Duan, Ryan Luo Li**
Association of Students for Research in Artificial Intelligence (ASTRA)
`astra.ai.lab@gmail.com`

## Abstract

Large language models (LLMs) have demonstrated potential in fact-checking claims, yet their capabilities in verifying claims in multilingual contexts remain largely understudied. This paper investigates the efficacy of various prompting techniques, viz. Zero-Shot, English Chain-of-Thought, Self-Consistency, and Cross-Lingual Prompting, in enhancing the fact-checking and claim-verification abilities of LLMs for Arabic claims. We utilize 771 Arabic claims sourced from the X-fact dataset to benchmark the performance of four LLMs. To the best of our knowledge, ours is the first study to benchmark the inherent Arabic fact-checking abilities of LLMs stemming from their knowledge of Arabic facts, using a variety of prompting methods. Our results reveal significant variations in accuracy across different prompting methods. Our findings suggest that Cross-Lingual Prompting outperforms other methods, leading to notable performance gains.

## 1 Introduction

Large language models (LLMs) have demonstrated remarkable proficiency in a wide range of tasks Minaee et al. [2024]. One particular area where LLMs have shown promising capabilities is in fact-checking and claim verification Choi and Ferrara [2024], Hoes et al. [2023], Lee et al. [2020], Zhang and Gao [2023]. The rise of fake news and misinformation in recent years has been well-documented, making fact-checking and claim verification essential to combat the rapid spread of misinformation.

However, previous work on fact-checking and claim verification using LLMs has primarily focused on English and Chinese facts and claims, leaving a significant gap in the exploration of multilingual fact-checking Cao et al. [2023], Quelle and Bovet [2024], Zhang et al. [2024]. This paper addresses this gap by focusing on fact-checking in Arabic, an inherently complex language due to its rich morphology, diverse dialects, and significant variation between written Modern Standard Arabic and spoken forms, using LLMs, which remains an under-explored domain. To this end, we benchmark LLM performance on a filtered dataset of 771 Arabic claims sampled from the X-fact dataset Gupta and Srikumar [2021a].

We utilize a variety of leading prompting techniques, including Zero-Shot (as a Baseline), English Chain-of-Thought Wei et al. [2023], Self-Consistency Wang et al. [2023], and Cross-Lingual Prompting Qin et al. [2023], to evaluate the effectiveness of LLMs in verifying Arabic claims. We present the variations in the accuracy of LLMs across different prompting methods. To our knowledge, this is the first work to evaluate the factual Arabic knowledge possessed by LLMs and their inherent Arabic fact-checking abilities based on this knowledge.

The remainder of this paper is organized as follows: In Section 2, we review related work. In Section 3, we define the problem of claim verification as explored in this paper. In Section 4, we describe the

---

\* Equal contribution

38th Conference on Neural Information Processing Systems (NeurIPS 2024).

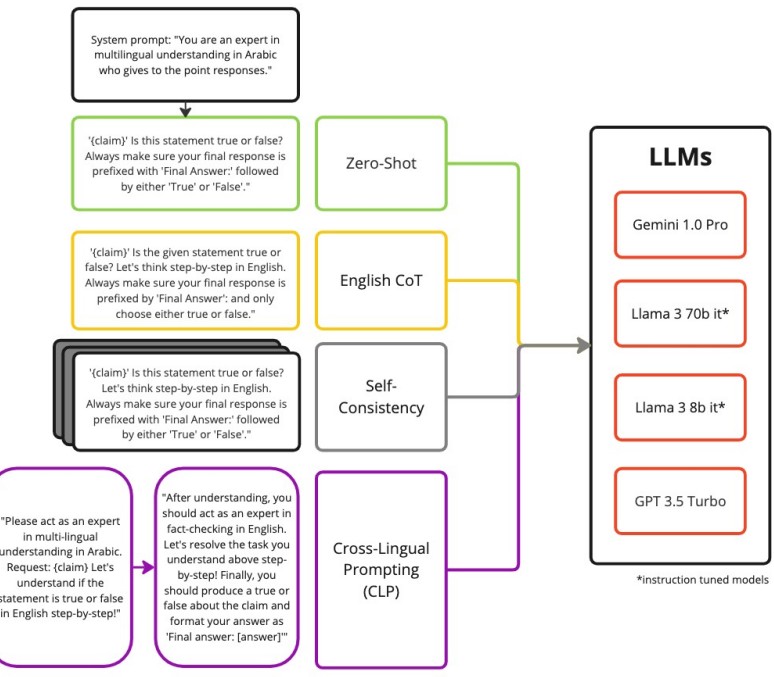

Figure 1: Workflow for comparing prompting strategies (Zero-Shot, English Chain-of-Thought (CoT), Self-Consistency, and Cross-Lingual Prompting (CLP)) used to evaluate the Arabic fact-checking capabilities of LLMs.

datasets, models, and evaluation methods used. We discuss our experiments in Section 5 and present our results in Section 6. Finally, we conclude in Section 7 and suggest directions for future research.

## 2 Related Work

**Fact-Checking using LLMs** With the rise of widespread misinformation, various studies have examined the capabilities of LLMs in fact-checking and claim verification. LLMs such as GPT-3 and GPT-4 excel in fact-checking when provided with sufficient contextual information, though they suffer from inconsistent accuracy Quelle and Bovet [2024]. Tian et al. 2023 suggests enhancing LLM factuality by fine-tuning models with automatically generated factuality preference rankings, which leads to improved factual accuracy without the need for human labeling. Cheung and Lam 2023 incorporates external evidence-retrieval to bolster fact-checking performance for the Llama model. Hu et al. 2023 examines the factual knowledge possessed by LLMs and their fact-checking capabilities using prompting techniques such as zero-shot, few-shot, and Chain-of-Thought.

**Multilingual Fact-Checking using LLMs** While there have been significant advancements in LLM-based fact-checking in English, multilingual fact-checking using LLMs remains relatively under-explored. Shafayat et al. 2024 examines the factual accuracy of LLMs across nine languages, including Arabic. Cekinel et al. 2024 explores cross-lingual learning and low-resource fine-tuning for fact-checking in Turkish, and uses in-context learning to evaluate LLMs' performance in this task.

**Arabic and LLMs** NLP in the Arabic language has seen significant advancements Darwish et al. [2021], Guellil et al. [2021] with Large Language Models (LLMs). Alyafeai et al. 2023 evaluates ChatGPT on a variety of Arabic NLP tasks. Pre-trained language models and language models fine-tuned on Arabic data have also demonstrated state-of-the-art performance in Arabic classification and generative tasks Alghamdi et al. [2023], Antoun et al. [2021], Deen et al. [2023]. Despite advancements in LLMs' capabilities in Arabic, fact-checking using LLMs remains under-explored.

| Claim | | Label |
|---|---|---|
| **Arabic** | **English Translation** | |
| وزيرة الصحة الفلسطينية تخرج عن طورها بسبب تفشي فايروس كورونا المستجد. | The Palestinian Minister of Health is out of her position due to the outbreak of the new Coronavirus. | 0 |
| طبيب مصري يقول إنّ مناعة التونسيين قد تكون علاجاً جديداً لفايروس كورونا (كوفيد-19). | An Egyptian doctor says that Tunisians' immunity may be a new treatment for the Coronavirus (COVID-19) | 0 |
| رئيس البرتغال يقف في المتجر وسط المواطنين ينتظر دوره. | The President of Portugal stands in the store among the citizens waiting for his turn | 1 |
| إصابة الفنانة رجاء الجداوي بفايروس كورونا المستجد(كوفيد_19) خلال تواجدها في مسقط رأسها بمحافظة الإسماعيلية | The artist, Ragaa Al-Jeddawi, was infected with the new Coronavirus (Covid_19) while she was in her home-town in Ismailia Governorate. | 1 |

Figure 2: Examples of Arabic claims, their English translations, and ground-truth labels (0: false; 1: true) from the test data.

Althabiti et al. 2024 present Ta'keed: an LLM-based system for explainable Arabic fact-checking, and achieve promising results. In this work, we benchmark the Arabic fact-checking abilities of several multilingual LLMs using a variety of prompting methods.

# 3 Problem Definition

We treat claim verification as a binary classification task. For each claim $x_i$ in our test dataset $\delta$ we prompt an LLM $l$ to classify the claim as either 'true' ($\hat{y} = 1$) or 'false' ($\hat{y} = 0$), where $\hat{y}$ is the value predicted by $l$. In the case that $l$ fails to return a binary value (inconclusive response) for $\hat{y}$, we take $\hat{y} = \neg y$.

# 4 Experimental Setup

## 4.1 Datasets

We utilize the X-fact dataset Gupta and Srikumar [2021a] as the source for the Arabic claims. The dataset is organized into several splits: Train, Development (Dev), In-domain Test ($\alpha_1$), Out-of-domain Test ($\alpha_2$), and Zero-Shot Test ($\alpha_3$). We filter out those claims whose ground truth labels differ from either 'true' or 'false' from the Train, Dev, and In-domain Test ($\alpha_1$) splits to create a test dataset $\delta$ containing 771 claims in Arabic:

$$\delta = \{(x_1, y_1), (x_2, y_2), ..., (x_n, y_n)\}$$

where $x_i$ is a claim in Arabic and $y_i \in \{0, 1\}$ is its ground truth label.

We note that 730 of the claims in the test dataset are false, while 41 are true. A sample from the test dataset is presented in Figure 2. Appendix A.1 contains further details about the test dataset.

## 4.2 Models

We conduct our experiments on Meta AI's Llama 3 8B and Llama 3 70B MetaAI [2024], Google DeepMind's Gemini 1.0 Pro Anil et al. [2023], and OpenAI's GPT-3.5-turbo. [2] For all models included in the study, we set the temperature to 0.7. The maximum possible token length for the outputs was set for each model given their respective context lengths.

---
[2] https://platform.openai.com/docs/models/gpt-3-5-turbo

### 4.3 Evaluation

We calculate an accuracy score for each LLM tested in each experiment. This accuracy score $s$ is expressed as a percentage value as follows:

$$s = \frac{n_c}{n} \times 100\%$$

where $n_c$ is the number of correct class predictions made by the LLM and $n$ is the size of the test dataset. As mentioned in Section 3, inconclusive responses are treated as incorrect classifications.

## 5 Experiments

Figure 1 depicts the four prompting techniques used.

**Zero-Shot Prompting** We employ zero-shot prompts to gauge the baseline performance of the LLMs on the test data. A zero-shot prompt simply contains an Arabic claim $x_i$ from the test dataset $\delta$ and an instruction $Z$ to classify the claim as either 'true' or 'false'. As such, the LLM $l$'s response is:

$$\hat{y} = l(x_i, Z)$$

**English Chain-of-Thought** Chain-of-Thought (CoT) prompting has been shown to significantly improve performance across various tasks Wei et al. [2023], including claim verification Hu et al. [2023]. This method enables models to articulate a clear, human-like, step-by-step reasoning process before arriving at a conclusion. Typically, in a zero-shot CoT prompt, the instruction "Let's think step by step" is added to the original instruction $Z$ to create a new instruction $Z_{\text{CoT}}$. The response $r_i$ of the LLM $l$ to an Arabic claim $x_i$ from the test dataset $\delta$ is computed as follows:

$$r_i = l(x_i, Z_{\text{CoT}})$$

$$r_i = (p_i, \hat{y}_i)$$

where $p_i$ represents the reasoning path followed by the language model to arrive at the final answer $\hat{y}_i$.

We explore English Chain-of-Thought Qin et al. [2023], i.e. we add the instruction "Let's think step-by-step in English" to the original instruction $Z$. Since the test data is in Arabic, we hypothesize that prompting the model to reason out the answer in English would increase the likelihood of the LLM understanding the Arabic claim, thereby leading to performance gains.

**Self-Consistency** Wang et al. 2023 shows that replacing the greedy decoding used in Chain-of-Thought with 'self-consistency' significantly improves CoT reasoning. Self-consistency involves prompting a language model to generate a variety of reasoning paths to arrive at an answer and marginalizing these reasoning paths to choose the most consistent answer as the final answer.

We add Self-Consistency to Cross-Lingual CoT. For an Arabic claim $x$, we prompt the LLMs to generate *three* reasoning paths in English and obtain three responses such that $r_i = (p_i, \hat{y}_i)$. We choose the most consistent value of $\hat{y}_i$ as the final answer.

**Cross-Lingual Prompting** Qin et al. 2023 leverage Cross-Lingual Prompting (CLP) to enhance zero-shot Chain-of-Thought reasoning in language models in multilingual settings. They show that CLP outperforms popular prompting techniques including English Chain-of-Thought.

CLP involves two steps: **(i)** Cross-Lingual Alignment Prompting, where the language model is prompted to understand the Arabic claim verification task step-by-step in English, and **(ii)** Task-specific Solver Prompting, where the language model is prompted to solve the task using CoT reasoning.

| Model | Correct | Incorrect | Inconclusive | Accuracy % | % Increase |
|---|---|---|---|---|---|
| Llama 3 8B-instruct | | | | | |
| Zero-Shot (Baseline) | 455 | 305 | 11 | 59.01 | — |
| English Chain-of-Thought | 500 | 209 | 38 | 66.93 | 13.42 |
| Self-Consistency | 529 | 201 | 41 | 68.61 | 16.27 |
| Cross-Lingual Prompting | 664 | 91 | 9 | **86.55** | 46.67 |
| Llama 3 70B-instruct | | | | | |
| Zero-Shot (Baseline) | 310 | 438 | 23 | 40.21 | — |
| English Chain-of-Thought | 472 | 265 | 34 | 61.22 | 52.25 |
| Self-Consistency | 460 | 247 | 64 | 59.66 | 48.37 |
| Cross-Lingual Prompting | 620 | 134 | 17 | **80.42** | 100.00 |
| Gemini 1.0 Pro | | | | | |
| Zero-Shot (Baseline) | 236 | 531 | 5 | 30.60 | — |
| English Chain-of-Thought | 383 | 307 | 81 | 49.68 | 62.35 |
| Self-Consistency | 405 | 322 | 44 | **52.53** | 71.67 |
| Cross-Lingual Prompting | 381 | 385 | 5 | 49.41 | 61.47 |
| GPT-3.5-turbo | | | | | |
| Zero-Shot (Baseline) | 468 | 279 | 21 | 60.94 | — |
| English Chain-of-Thought | 461 | 244 | 66 | 59.79 | -1.89 |
| Self-Consistency | 491 | 235 | 45 | 63.68 | 4.50 |
| Cross-Lingual Prompting | 603 | 116 | 2 | **78.21** | 28.34 |

Table 1: Results for each prompting method and LLM. '% Increase' denotes the percentage increase in model performance from the baseline (zero-shot).

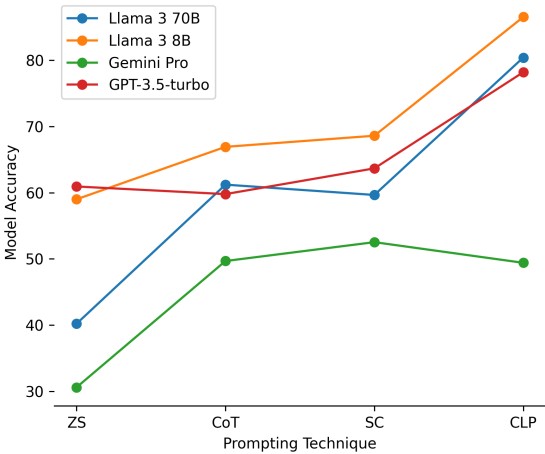

Figure 3: Model Accuracy vs Prompting Method

## 6 Results and Analysis

Our findings for each prompting approach are presented in Table 1. Figure 3 shows the relation between the prompting technique and model accuracy for each model. The percentage increase in accuracy from the baseline for each prompting method and model is shown in Figure 4. Generally, we find that the model accuracy increases from zero-shot to Cross-Lingual CoT to Self-Consistency, and typically reaches its maximum value in the CLP setting.

Figure 6 shows the relation between the prompting technique and the number of inconclusive answers for each LLM. As shown in the figure, the number of inconclusive responses, on average, increases when going from zero-shot to Cross-Lingual CoT or Self-Consistency. This number decreases in the CLP setting, in which the fewest inconclusive responses are returned.

Figure 5 shows a mostly linear relationship between the prompting technique and the number of correct answers for each LLM.

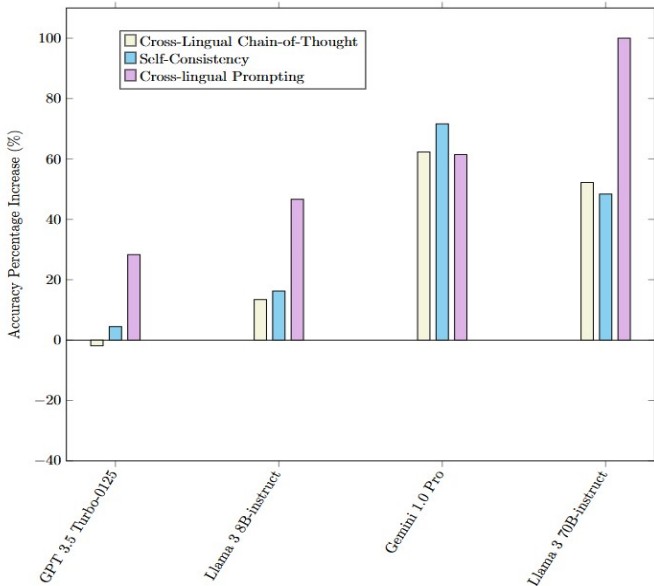

Figure 4: Percentage Increase from the Baseline (Zero-Shot) for each Prompting Method and LLM.

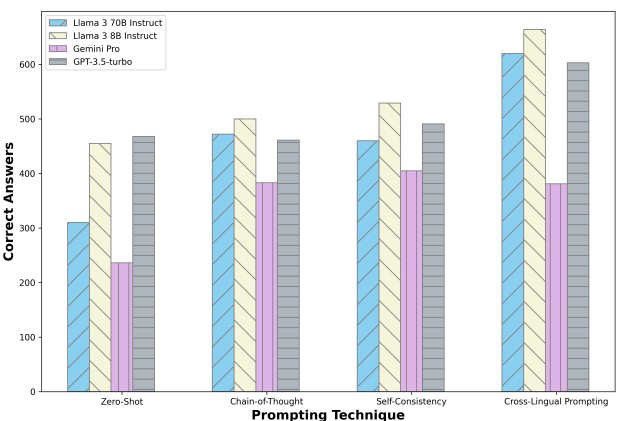

Figure 5: Variation of the number of correct answers with prompting method for each model.

## 6.1 Zero-Shot

**Accuracy** We find that Llama 3 70B Instruct achieves an accuracy of 40.21%, and Llama 3 8B achieves a higher accuracy of 59.01%. GPT-3.5-turbo achieves the second-best accuracy of 60.94% while Gemini Pro performs the worst with an accuracy of 30.60%.

**Inconclusive Responses** The language models show varying levels of inconclusive responses, with Llama 3 70B, Llama 3 8B, and GPT-3.5-turbo recording 23, 11, and 21 inconclusive responses respectively. Interestingly, despite a lower overall accuracy, Gemini 1.0 Pro returns only 5 inconclusive responses, which could indicate a propensity to deliver more decisive answers, albeit incorrect.

We observe that in the zero-shot setting, the LLMs are not effective fact-checkers and have room for improvement.

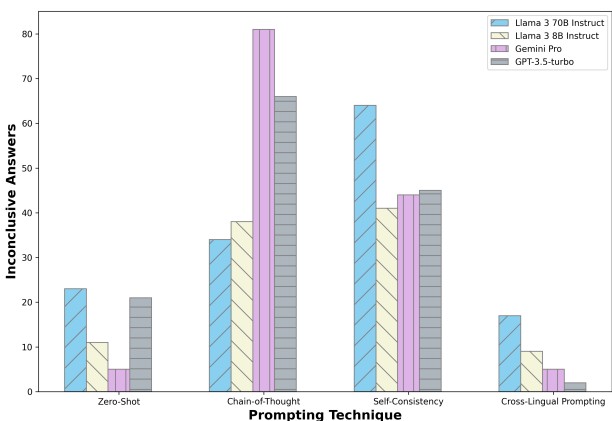

Figure 6: Variation of inconclusive answers for each model with different prompting techniques.

## 6.2 English Chain-of-Thought

**Accuracy** We observe that the English Chain-of-Thought (CoT) approach generally improves accuracy across most models compared to the zero-shot baseline. Llama 3 70B Instruct's accuracy increases by 52.25% (from 40.21% to 61.22%) in the CoT setting. Llama 3 8B Instruct's accuracy increases from 59.01% to 66.93%, a 13.42% increase. Gemini Pro's performance rises by 62.35% (49.68% from 30.60%).

In contrast, GPT-3.5-turbo performs with similar accuracy in the Cross-Lingual CoT setup, with a 1.89% drop in accuracy from its zero-shot performance.

**Inconclusive Responses** Despite the increase in accuracy for most LLMs, there was a significant rise in inconclusive responses across all models when applying the Cross-Lingual CoT method. This was particularly marked in Gemini Pro and GPT-3.5-turbo where inconclusive responses shot up to 61, 81, and 66 respectively. We find that while Cross-Lingual CoT appears to improve accuracy by allowing the LLMs to reason out the answers in English, it also seems to introduce greater uncertainty, leading to a higher number of inconclusive responses.

We find that generally, while English Chain-of-Thought leads to a rise in the number of inconclusive responses, the LLMs mostly return more correct answers, leading to a net increase in accuracy.

## 6.3 Self-Consistency

**Accuracy** We find that implementing Cross-Lingual CoT with Self-Consistency enhances model performance beyond Cross-Lingual CoT. For Llama 3 8B Instruct and Llama 3 70B Instruct, the accuracy increases by 16.27% and 48.37%, respectively. Gemini Pro's accuracy rises significantly, by 71.67%. GPT-3.5-turbo's accuracy increases by 4.50%. Llama 3 70B Instruct performs worse in the Self-Consistency setting than in the Cross-Lingual CoT setting.

**Inconclusive Responses** As shown in Figure 6, Self-Consistency leads to the highest number of inconclusive responses out of all the prompting methods. Llama 3 70B Instruct returns the highest number of inconclusive responses (64). We hypothesize that because the model is prompted to generate three lines of reasoning, it is susceptible to hallucinations and indeterminate chains of thought.

We observe that integrating Self-Consistency with Cross-Lingual CoT leads to an increase in the number of inconclusive responses returned by the LLMs. However, due to a rise in the number of correct answers, there is a net increase in model accuracy.

### 6.4  Cross-Lingual Prompting

**Accuracy** We find that cross-lingual prompting (CLP) often leads to the best model performance out of all the four prompting techniques. Llama 3 8B Instruct's accuracy improves by 46.67% over the baseline to achieve an accuracy of 86.55%, the highest among all tested models and methods. Similarly, GPT-3.5-turbo's performance also benefits from CLP, with its accuracy rising to 78.21% from a baseline of 60.94%. Llama 3 70B's performance reaches 80.42% from its baseline of 40.21%, a 100% improvement.

**Inconclusive Responses** Interestingly, while CLP improved accuracy across the board, it also led to a reduction in inconclusive responses for most models, indicating an increase in decisiveness. We observe a reduction in inconclusive responses from 11 to 9 for Llama 3 8B, 23 to 17 for Llama 3 70B, and 21 to 2 for GPT-3.5-turbo from zero-shot to CLP. The number of inconclusive responses remains unchanged for Gemini Pro.

Our findings suggest that CLP is extremely effective in clarifying the decision-making processes for these LLMs in an Arabic context while maintaining accuracy.

## 7  Conclusion and Future Work

In this study, we examined the Arabic fact-checking and claim verification capabilities of four LLMs: Llama 3 8B Instruct, Llama 3 70B Instruct, Gemini 1.0 Pro, and GPT-3.5-turbo. We employed four prompting techniques: Zero-Shot, English Chain-of-Thought, Self-Consistency, and Cross-Lingual Prompting. Our findings reveal that although these LLMs perform inadequately in a zero-shot setting, prompting techniques that engage reasoning capabilities significantly enhance their performance. In particular, Cross-Lingual Prompting showed substantial improvement in accuracy, suggesting that leveraging the reasoning capabilities of LLMs through sophisticated prompting strategies can effectively address the challenges posed by the complex morphology and diverse dialects of the Arabic language.

In future work, we aim to expand our dataset to establish a comprehensive benchmark for Arabic claim verification that includes diverse claims from various domains. Additionally, a future study could investigate how LLMs perform on fact-checking for claims in various independent Arabic dialects. Given the promising results of Cross-Lingual Prompting, we plan to explore other advanced prompting strategies, including few-shot prompting and Cross-Lingual Prompting with Self-Consistency, to further enhance performance.

## Limitations

The scope of our analysis is restricted to a select group of LLMs. It would be interesting to investigate the Arabic fact-checking abilities of other leading models such as OpenAI's GPT-4 and Anthropic's Claude 3 series. Additionally, our dataset mainly comprises claims labeled as ground-truth false (730) as opposed to true (41). While this skew does not compromise the assessment of the LLMs' verification abilities, a more balanced distribution could provide deeper insights into their fact-checking capabilities in Arabic.

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

# A Appendix

## A.1 Dataset Creation

## A.2 Dataset Statistics

The X-fact dataset Gupta and Srikumar [2021b] was utilized as our primary data source. The claims in the dataset are sourced from https://misbar.com.

## A.3 Preprocessing Steps

**1. Filtering:** We filtered the dataset to include only claims that were labeled as either "true" or "false". Claims with other labels or those lacking verification were excluded from the finalized dataset.
**2. Combining Splits:** After filtering, the claims from the Train, Dev, and In-domain Test ($\alpha_1$) splits were combined to form a single dataset for our experiments.

## A.4 Dataset Composition

Table 2 shows the total number of Arabic claims and the number of Arabic claims filtered. After pre-processing, the test dataset contained a total of 771 Arabic claims.

Number of claims from Train set: 643

Number of claims from Dev set: 88

Number of claims from In-domain Test ($\alpha_1$) set: 40

## A.5 Label Distribution

TRUE Claims: 41 claims (5.32%)
FALSE Claims: 730 claims (94.68%)

| Dataset Split | Total Number of Claims | Filtered Number of Arabic Claims (True & False) |
|---|---|---|
| Train | 18246 | 643 |
| Dev | 3657 | 88 |
| In-domain Test ($\alpha_1$) | 2406 | 40 |
| **Total** | **24309** | **771** |

Table 2: Summary of the dataset splits before and after filtering claims labeled as 'TRUE' or 'FALSE'.

**Compute Resources**

All experiments were conducted using a combination of cloud-based GPU instances and local compute resources. The specific details of the compute setup are outlined below:

**GPU Resources**

For training and evaluating the LLMs, we utilized the following GPU configurations:

- **Cloud GPU Instances:** Experiments were primarily conducted on NVIDIA A100 40GB GPUs hosted on cloud providers (e.g., AWS EC2, Google Cloud Platform). Each instance included 8 A100 GPUs with 320GB of total VRAM. The experiments on these instances ran across multiple GPUs in parallel for faster throughput.
- **Local GPU Instances:** Some experiments were run locally on a system equipped with 2 NVIDIA RTX 3090 GPUs, each with 24GB of VRAM.

**Compute Time**

- **Zero-Shot Prompting:** Each model required approximately 1 hour of compute time on a single GPU for evaluating the 771 claims using zero-shot prompting.
- **Chain-of-Thought Prompting:** English Chain-of-Thought and Cross-Lingual Chain-of-Thought evaluations required about 3 hours per model per experiment, as generating reasoning chains increased compute time.
- **Self-Consistency:** The self-consistency experiments, which required generating multiple reasoning paths for each claim, took approximately 6 hours per model.

**Total Compute Resources**

The total compute time across all models and experiments was approximately 100 GPU hours. Most of this time was spent on the Self-Consistency and Cross-Lingual Prompting experiments due to the additional reasoning paths generated.

**Memory and Storage**

Each experiment required at least 200GB of storage for caching intermediate results and model checkpoints. The average memory usage was 120GB during peak execution of the larger models (e.g., Llama 3 70B).

**Software Environment**

All experiments were run using the following software stack:

- **Operating System:** Ubuntu 20.04 LTS
- **Deep Learning Framework:** PyTorch 2.0
- **CUDA Version:** 11.7
- **Other Dependencies:** Transformers (Hugging Face), Python 3.9, and specific drivers for NVIDIA GPUs.

