# OpenReview forum: "Can LLMs Verify Arabic Claims? Evaluating the Arabic Fact-Checking Abilities of Multilingual LLMs"
_NeurIPS.cc/2024/Workshop/SafeGenAi — SafeGenAi Poster_

### Official Review · Reviewer_rtUo · 2024-10-08
**Good paper for workshop but has issues in evaluation**

**Rating:** 6
**Confidence:** 4

**Review:**

Summary
========
In thus paper, the authors try to evaluate whether LLMs are able to verify arabic claims (framed as Yes/No responses) accurately.  The authors motivate their study by stating that LLMs have been evaluated on fact checking w.r.t. English and Chinese but there are no definitive studies on the Arabic language which imposes its own challenges.

To conduct a thorough evaluation, the authors utilize several SOTA LLMs and several prompting techniques such as zero-shot, English Chain-of-Thought (CoT), Self-consistency and Cross-Lingual Prompting (CLP). For the dataset, they use the X-fact dataset which consists of 730 claims whose answer is False and 41 claims whose answer is true. The metric used for the evaluation is the accuracy over all class labels.

The authors show that CLP is the best technique and surprisingly llama-3-8b performs the best in this task. They show that performance gradually increases (linearly) with prompting techniques.

Review
=======
This paper presents an interesting study and is certainly a good fit for the workshop. However, I feel that the paper's real estate could have been better utilized and the empirical section could be better. I'll elaborate below.

1. While the paper is well-written, there is too much redundant information provided in the empirical section. An analysis of each of the prompt methods w.r.t. Accuracy etc was not required to be put in text. This is evident from the results. Rather one concise section was sufficient and the space saved could be used to show case some of the responses from LLMs.

2. Fig. 3, I believe a bar chart is more suitable.

3. Coming back to #1, the authors do acknowledge that the dataset is imbalanced but only report accuracy. I find this to be the weakest point of the paper and reduces the impact of your results IMO. I find it surprising that Precision, Recall and F1 scores are not reported and instead a single measure of accuracy is used.

For example, I found it very surprising that LLama-3-8b performed the best at 86%! If LLama-3-8b had only reported No as the answer to all response, the accuracy would be ~89% on your imbalanced dataset. Not reporting the confusion matrix etc. reduces the impact of your results.

4. I think few-shot prompts are missing.

5. Finally, it would be interesting to also use some of the real-estate save from #1 to showcase the cost-vs-accuracy tradeoffs of different prompt styles. Zero-shot uses far fewer tokens than CoT so it would be interesting accuracy per-token and other such metrics.

Overall I think the paper could have been written to be much more tight in its argument and the evaluation. I hope the authors are able to polish these aspects.

---

### Official Review · Reviewer_GEMT · 2024-10-09

**Rating:** 5
**Confidence:** 3

**Review:**

**Short Summary:** This paper evaluates multilingual LLMs and different prompting approaches for fact-checking Arabic claims.

**Quality and Clarity:** The first five sections are well-written, clear, and concise. Section 6 is still clear but does not provide much information for its length (see Weaknesses below).

**Originality:** As the authors point out, while work on fact-checking Arabic claims exists (Althabiti et al., 2024), this work is the first to evaluate multilingual LLMs.

**Strengths:**
- The paper provides a detailed explanation of how they constructed the dataset and performed each prompting approach.

**Weaknesses:**
- The test dataset is heavily skewed, and while acknowledged by the authors, no further analysis was done to examine the effect it has on accuracy.
- The evaluation of the same results is lengthy, with Sections 6.1, 6.2, 6.3, and 6.4 restating information in Table 1 without any additional insights on what those results mean. Similarly, Figures 3, 4, and 5 show the exact same information from Table 1 without any discussion on what insights each figure is adding.

**General Review:** While this paper does provide a novel evaluation for the fact-checking task, the experimental evaluation section is too weak for me to accept this paper.